# Diagnostic performance of the AID line probe assay in the detection of *Mycobacterium tuberculosis* and drug resistance in Romanian patients with presumed TB

**Andrea Rachow**[1,2]*, **Elmar Saathoff**[1,2], **Roxana Mindru**[3], **Oana Popescu**[3], **Doinita Lugoji**[3], **Beatrice Mahler**[3], **Matthias Merker**[4], **Stefan Niemann**[4], **Ioana D. Olaru**[4,5], **Sabine Kastner**[1], **Michael Hoelscher**[1,2], **Christoph Lange**[4,5,6,7], **Elmira Ibraim**[3]

**1** Division of Infectious Diseases and Tropical Medicine, University Hospital, LMU Munich, Munich, Germany, **2** German Centre for Infection Research (DZIF), Munich, Germany, **3** Marius Nasta Institute of Pulmonology, Bucharest, Romania, **4** Division of Clinical Infectious Diseases, Research Center Borstel, Borstel, Germany, **5** German Centre for Infection Research (DZIF), Hamburg-Lübeck-Borstel-Riems, Hamburg, Germany, **6** Respiratory Medicine & International Health, University of Lübeck, lübeck, Germany, **7** Baylor College of Medicine and Texas Children´s Hospital, Houston, TX, United States of America

* rachow@lrz.uni-muenchen.de

**Data Availability Statement:** Anonymized, complete data are available in the Supporting Information.

## Abstract

### Background

The AID line probe assay has shown promising evaluation data on the detection of *Mycobacterium tuberculosis* as well as 1st- and 2nd-line drug resistance, using isolates and selected clinical samples in previous studies.

### Methods

The diagnostic performance of three AID-modules (AID INH/RIF, AID FQ/EMB and AID AG) was analyzed in sputum samples from patients with presumed tuberculosis against culture methods and phenotypic drug resistance as reference standards.

### Results

59 patients had culture-confirmed tuberculosis. All AID modules showed moderate sensitivity (46/59, 78.0%, 65.3–87.7) and very good specificity (100%, 95.5%, 93.7%). There was a high proportion of invalid tests, resulting in 32.6%, 78.3% and 19.6% of 46 AID-positive tuberculosis cases, who could not be assessed for drug resistance by the AID INH/RIF-, AID FQ/EM- and AID AG-module, respectively. A small number of patients showed drug resistance by reference standards: Three MDR-TB cases plus three, one and one patients with resistance to streptomycin, fluoroquinolones and aminoglycosides, respectively. The AID-assay detected all MDR-TB cases, two of three streptomycin-resistant TB cases, one of one of fluoroquinolone-resistant and missed one aminoglycoside-resistant TB case.

**Funding:** The study was funded through the German Center for Infection Research (DZIF), TTU 02.804. The AID tests as well as equipment and reagents needed for the conduct of AID-tests were provided by the AID Diagnostika GmbH, Ebinger Strasse 4, D-72479 Straßberg. The AID Diagnostika GmbH was not involved in study design, data analysis and writing of this manuscript.

**Competing interests:** The authors have declared that no competing interests exist.

## Discussion

The high proportion of invalid results precludes the use of the AID-assay from direct sputum-based tuberculosis and drug-resistance testing.

## Introduction

Tuberculosis (TB) is still a major cause of morbidity and mortality worldwide [1]. In 2019, there were an estimated 1.2 million deaths due to TB and about 10 million new TB cases [2]. Within the European Union, Romania has a relatively high number of TB cases, with an estimated incidence of 66 per 100,000 people in 2019, and 13,000 (range 11,000–15,000) new and relapse cases per year [3]. Within a century of their discovery, antibiotics are becoming less effective due to the increase in antibiotic resistance [4]. Among the infections most affected by resistance development is TB. Worldwide, in 2019, 465,000 (range 400,000–535,000) people were estimated to be actively infected with rifampicin resistant *Mycobacterium tuberculosis* (*M. tuberculosis*) with 79% having multidrug-resistant TB (MDR-TB) [2], which includes the additional resistance against isoniazid, another highly potent drug. Finally, about 20.1% of MDR strains of *M. tuberculosis* were also resistant to fluoroquinolones [2]. Among the patients with MDR-TB, who started their treatment in 2017, only 57% completed it successfully and about 15% died [2]. In order to stop the evolution of drug-resistant strains of *M. tuberculosis* and their further spread within communities and to achieve the optimal treatment outcome for each TB patient, the early initiation of a correct and effective antibiotic treatment is key. This, however, requires an immediate and accurate detection of drug resistance mutations. In TB patients with shown or presumed rifampicin-resistant TB (RR-TB), WHO recommends the use of commercial line probe assays (LPAs) for a further assessment of prevalent first and second line drug resistance mutations [5]. LPAs are based on genotypic methods and one of their advantages is that results are available within one day. In addition, LPAs have the potential for high throughput testing and reduced biosafety requirements as compared to phenotypic resistance testing. There is a new commercial LPA available, manufactured by AID Autoimmun Diagnostika GmbH, Strassberg, Germany [6], which includes the detection of 1st and 2nd line drug resistance. Previous studies showed the assay to be an accurate tool for detection of drug-resistance in various stored acid fast bacilli (AFB)-positive specimens and in clinical isolates [6,7]. In this clinical diagnostic evaluation study from Bucharest, Romania, we prospectively assessed the performance of three different AID LPA modules in the detection of *M. tuberculosis* as well as first and second line drug resistance directly on sputum samples from outpatients with signs and symptoms of active pulmonary TB.

## Methods

### Study design and data collection

We performed a prospective, diagnostic evaluation study in symptomatic adult patients who consecutively presented to the pulmonology outpatient department of the Marius-Nasta-Institute (MNI) in Bucharest, Romania. Eligible patients had a chest radiograph compatible with pulmonary TB plus one of the following signs and symptoms indicative for TB: productive cough for more than two weeks, haemoptysis, fever, night sweats or substantial involuntary weight loss (refer to supplement for complete list of inclusion and exclusion criteria).

At the enrolment visit, a physical examination was performed, chest x-ray data were captured and a questionnaire was applied for clinical data collection. One follow up visit was

performed at eight weeks after enrolment in order to confirm whether or not a study participant was diagnosed with TB and on what grounds (microbiological versus clinical TB diagnosis).

## Sample processing and M. tuberculosis detection

At enrolment, two spontaneous sputum samples were collected from each study participant. Sample processing and analyses were performed at the National Reference Laboratory at the MNI. All sputum samples were decontaminated using the N-acetyl-L-cysteine-sodium hydroxide method [8] and were subsequently examined by microscopy and culture methods (refer to **supplement** for microbiological methods). Cultures which tested positive for *M. tuberculosis* were shipped to the National Reference Center for Mycobacteria at the Research Center Borstel, Germany for DNA extraction using the CTAB protocol as described previously and subsequent whole genome sequencing analysis [9], (see below).

## Phenotypic drug resistance testing

For culture based phenotypic 1st and 2nd line drug susceptibility testing (DST), every culture positive strain was inoculated on solid LJ media supplemented with rifampicin (40 mg/L), isoniazid (0.2 mg/L), streptomycin (4 mg/L), ethambutol (2 mg/L), amikacin (30 mg/L), kanamycin (30 mg/L), capreomycin (40 mg/L), ofloxacin (2 mg/L), or ethionamide (40 mg/L) [10,11].

## AID assay performance and reader rules

For AID assay evaluation, DNA was extracted from one decontaminated sputum sample of each participant using a commercially available extraction kit, HAIN GenoLyse VER 1.0. Each of the three investigated AID LPA modules includes the detection of *M. tuberculosis* as well as the respective drug resistance evaluation. The AID RIF/INH module detects mutations against rifampicin and isoniazid at the -16, -15, and -8 *inhA* or the S315T KatG aminoacid locus and mutations in the codons 516, 526 and 531 of the *rpoB* gene. Module 2, AID AG investigated the aminoglycosides streptomycin, amikacin and capreomycin (CAP) aminoacid exchanges at positions 43 and 88 of RpsL and gene mutations at 513, 514, 515, 517 in *rrs* for streptomycin resistance, and at positions 1401, 1402 and 1484 in *rrs* for amikacin/capreomycin resistance. Module 3, AID FQ/EMB, evaluates fluoroquinolone resistance resulting from aminoacid exchanges at positions 90, 91 and 94 of GyrA and ethambutol resistance resulting from aminoacid exchanges at positions 306 and 918 of EmbB. The assay was run according to the instructions of the manufacturer. *M. tuberculosis* strain ATCC 25177 was used as positive control, molecular-grade water as negative control. In case of any invalid result obtained, i.e. when reaction zones for either conjugation control, for wildtype and/or point mutations are missing or incomplete, the same sample was tested once again by the respective AID module. The AID hybridization results were read by two independent readers (refer to **supplement** for reader rules), who were blinded to all other lab results and clinical data, one Romanian, one German staff member (MNI and LMU, respectively). In case of ambiguous results, a third reader, blinded to the first two reader results, was consulted.

## Whole genome sequencing and drug resistance prediction

WGS was performed with Illumina technology (NextSeq 500), using Nextera XT library preparation kit, and reaching at least 50x average coverage of the reference genome (NC_000962.3). Fastq files were processed with the MTBseq pipeline as reported previously, and hosted at the European Nucleotide Archive under the accession number PRJEB42600 [12]. Variants (single

nucleotide polymorphisms (SNPs), insertions and deletions (InDels)) were called with a minimum coverage of 4 reads in both forward and reverse orientation, 4 reads calling the allele with at least a Phred score of 20, and 75% allele frequency. We investigated the following genes: katG, inhA, fabG1, fabG1 promotor region, and ndh for isoniazid resistance; rpoB for rifampicin resistance; rpsL, rrs, gidB, whiB7 promotor region for streptomaycin resistance; aftA embC embA embB for ethambutol resistance; gyrA, gyrB for fluoroquinolone resistance; rrs for amikacin resistance; rrs, tlyA for capreomycin resistance. Mutations present in the WHO technical guidelines were interpreted as resistance determining [13].

## Data analysis

The statistics software Stata/SE (version 15, StataCorp, College Station, Texas, USA) was used for data analysis. The analysis of the diagnostic performance of the three modules of the AID assay included the capacity to in a first step detect *M. tuberculosis* and in a second step to also detect mutations leading to drug resistance. For both analyses culture data were used as reference standard. For analysis purpose, all participants were allocated to one of the following four groups: A: Microbiologically confirmed TB, B: No TB, C: Clinical TB and D: Indeterminate TB status (**Fig 1**, also refer to online supplement for case definitions). Assay performance (sensitivity, specificity, positive and negative predictive value and agreement/area under the curve) was calculated using 2 by 2 tables. All performance estimates are accompanied by exact (Clopper-Pearson) confidence intervals.

## Ethics

This study was approved by the Ethics Committees of the MNI in Bucharest (registration number: 13143), the University of Luebeck (15–093) and the University of Munich (registration number: 38–15).

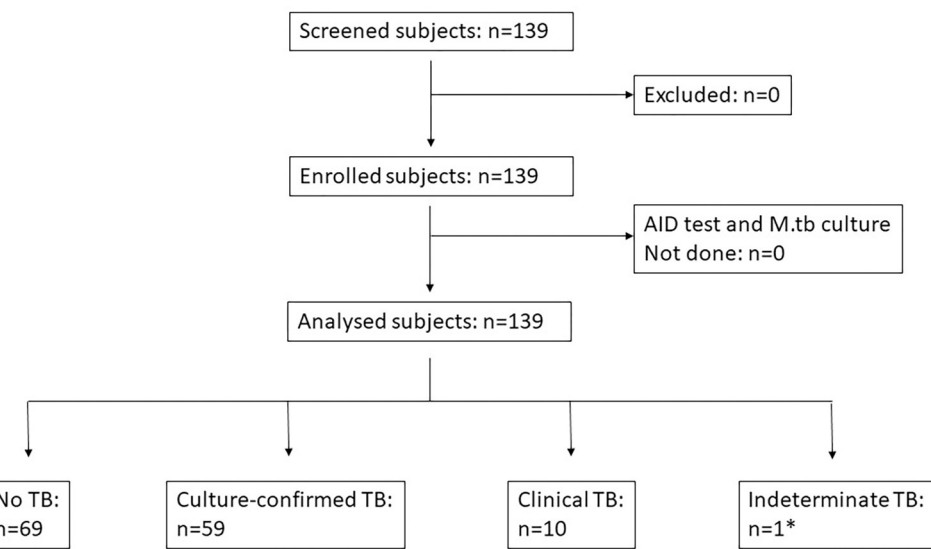

**Fig 1. Study flow-chart.** No TB: no growth of *M. tuberculosis* in any sputum sample and clinical recovery without anti-TB treatment; culture-confirmed TB: at least one *M. tuberculosis* positive culture result (either on LJ or VersaTEK liquid culture) for at least one out of two analyzed sputum samples, clinical TB: no microbiological proof of *M. tuberculosis*, however, anti-TB treatment was initiated based on clinical grounds; indeterminate TB: no fit to any of the other categories, i.e. *one patient, who was diagnosed with COPD and successfully treated with ciprofloxacin and showed contamination in all *M. tuberculosis*- sputum cultures.

## Results

### Study population

From March 2015 to April 2016, 139 adult patients with presumed pulmonary TB were screened for study eligibility and all of them were enrolled into the study (**Fig 1**). The demographic and clinical characteristics of the study population are shown in **Table 1**. No participant was lost-to-follow-up or died during the 8 weeks of follow-up.

### TB diagnosis by standard methods

Two sputum samples were collected from all 139 enrolled participants, resulting in 278 available culture results for liquid and solid culture, respectively. In total, 59/139 (42.4%) patients were diagnosed with microbiologically confirmed TB (**Fig 1**), and 43/139 (30.9%) were also positive by sputum smear microscopy, resulting in a sensitivity of 72.9% (43/59; 95%CI: 59.7–83.6) for smear microscopy **Table 2A**. In 69 participants with *M. tuberculosis* negative cultures and clinical recovery without anti-mycobacterial treatment, active pulmonary TB was excluded, (**Fig 1**). In 10 participants, all sputum cultures were *M. tuberculosis* negative, however, based on clinical findings (mostly radiology findings) and the medical history, the local clinical team initiated TB treatment, (**Fig 1**).

### M. tuberculosis detection by AID LPA

The sensitivity for *M. tuberculosis* detection of AID-LPA tests was 78.0% (95%CI: 65.3–87.7%) for the INH/RIF- and FQ/EMB- modules and 79.3% (66.6–88.8%) for the AG-module, respectively (**Table 2A**). As expected, sensitivity of all AID modules was much higher (93%, 80.9–98.5%) in patients with detectable acid-fast bacilli on smear microscopy (**Table 2B**) while it was quite low with about 37.5% (15.2–64.6%) for INH/RIF and FQ/EMB and 43.8% (9.8–70.1%) for AG in patients in whom no bacilli were microscopically visible (**Table 2B**). In general, there was a high specificity in all modules of 93.7% (84.5–98.2%) for the AG module to 97.1% (89.9–99.6%) for the FQ/EMB module (**Table 2A**). For one participant with confirmed HIV-coinfection, concordant positive results were obtained for all TB diagnostic tests performed in this study, including the three AID-LPA modules. All participants with clinical TB diagnosis had a *M. tuberculosis* negative result in all AID-LPA modules, except one participant with an invalid AG-module test result.

### Drug resistance testing

The assessment of drug resistance by the AID-LPA modules was characterized by a high number of false *M. tuberculosis* -negative and invalid AID-LPA results, (**Tables 3 and 4**).

Among those participants with an available and valid DST and AID-LPA result, the sensitivity was 100% for detection of mutations conferring resistances to rifampicin (95%CI: 29.2–100.0%) and isoniazid (95%CI: 15.8–100.0%), while specificity was 100% (88.1–100.0%) for detection of rifampicin-resistance and 96.7% (82.8–99.9%) for isoniazid-resistance, due to one additionally detected resistance to isoniazid compared to culture DST (Table 3). The detected rifampicin-resistances based on a mutation at gene locus *rpoB* 531 were confirmed for two participants by the sequencing data (**Table 4**), while the sequencing data were missing for the third participant, (**Table 4**). Likewise, two isoniazid-resistances (SNP S315T in KatG) detected by the AID-LPA were confirmed by sequencing, including that one which reported as isoniazid-sensitive based on culture DST. For a third INH-resistant strain found by DST and AID-LPA (SNP S315T in KatG and inhA -16, -15, -8), the results could not be confirmed by sequencing due to missing data. There was a fourth patient who showed resistance to isoniazid

**Table 1. Characteristics of study participants.**

| Variable | Categories | Distribution ALL, n/139 (col. %) | Distribution TB-pos. only, n/59 (col. %) |
|---|---|---|---|
| Age (years) | | | |
| | Mean (SD) | 48.19 (16.3) | 40.85 (14.53) |
| Age (years) | | | |
| | 19 to 40 | 52 (37.41) | 32 (54.24) |
| | 41 to 60 | 56 (40.29) | 23 (38.98) |
| | >/ = 60 | 31 (22.30) | 4 (6.78) |
| Sex | | | |
| | Female | 42 (30.22) | 17 (28.81) |
| | Male | 97 (69.78) | 42 (71.19) |
| HIV-status | | | |
| | Positive | 1 (0.72) | 1 (1.69) |
| | Negative | 138 (99.28) | 58 (98.31) |
| Location of residence | | | |
| | Urban | 106 (76.26) | 42 (71.19) |
| | Rural | 33 (23.74) | 17 (28.81) |
| Country of birth | | | |
| | Romania | 138 (99.28) | 58 (98.31) |
| | Other | 1 (0.72) | 1 (1.69) |
| Education | | | |
| | None | 1 (0.72) | 0 (0.0) |
| | Primary/vocational | 49 (35.25) | 16 (27.11) |
| | Secondary | 64 (46.04) | 30 (50.84) |
| | University and higher | 25 (17.99) | 13 (22.03) |
| Housing situation | | | |
| | Homeless | 2 (1.44) | 0 (0.0) |
| | Living with relatives/friends | 53 (38.13) | 32 (54.24) |
| | Renting accommodation | 17 (12.23) | 7 (11.86) |
| | Owning accommodation | 67 (48.20) | 20 (33.90) |
| Smoking (ever) | | | |
| | Yes | 101 (72.66) | 48 (81.36) |
| | No | 38 (27.34) | 11 (18.64) |
| Currently smoking | | | |
| | Yes | 76 (54.68) | 38 (64.41) |
| | No | 25 (17.99) | 10 (16.95) |
| | Missing | 38 (27.34) | 11 (18.64) |
| Alcohol (ever) | | | |
| | Yes | 100 (71.94) | 44 (74.58) |
| | No | 39 (28.06) | 15 (25.42) |
| Current alcohol consumption | | | |
| | Yes | 91 (65.47) | 42 (71.19) |
| | No | 10 (7.19) | 2 (3.39) |
| | Missing | 38 (27.34) | 15 (25.42) |
| Previous TB | | | |
| | Yes | 41 (29.50) | 7 (11.86) |
| | No | 98 (70.50) | 52 (88.14) |
| Cavities present on cxr | | | |
| | Yes | 40 (28.78) | 26 (44.07) |

(*Continued*)

**Table 1.** (Continued)

| Variable | Categories | Distribution ALL, n/139 (col. %) | Distribution TB-pos. only, n/59 (col. %) |
|---|---|---|---|
| | No | 96 (69.06) | 33 (55.93) |
| | Missing | 3 (2.16) | 0 (0.0) |
| % of lung involved on cxr | | | |
| | Mean (SD) | 27.12 (19.69) | 33.29 (21.69) |
| Ralph score* | | | |
| | Mean (SD) | 38.38 (31.81) | 51.00 (36.47) |
| Ralph score* | | | |
| | 5 to <40 | 83 (59.71) | 28 (47.46) |
| | 40 to <80 | 36 (25.90) | 15 (25.42) |
| | >80 to 130 | 20 (14.39) | 16 (27.12) |
| Pulmonary comorbidities | | | |
| | COPD | 22 (15.83) | 2 (3.39) |
| | Asthma | 4 (2.88) | 1 (1.69) |
| | Silicosis | 1 (0.72) | 1 (1.69) |
| Other comorbidities | Diabetes mellitus | 7 (5.04) | 1 (1.69) |

*reference: Ralph et al. [14], % = percentage, SD = Standard Deviation, n = number of participants in each stratum, cxr = chest x-ray, TB-pos. = TB-positive.

**Table 2. A: Diagnostic performance of smear microscopy and different AID modules for detection of *M. tuberculosis*.** B: Diagnostic sensitivity of different AID modules in participants with culture-confirmed TB, stratified by smear-microscopy status.

| Sub-group/test | Sensitivity | Specificity**** | Pos. pred. value | Neg. pred. value | Agreement/AUC |
|---|---|---|---|---|---|
| Smear-microscopy | 43/59 (72.9%; 95%CI: 59.7–83.6) | 69/69 (100%; 94.8–100) | 43/43 (100%; 91.8–100) | 69/85 (81.2%; 71.2–88.8) | 112/128 (87.5%; 80.5–92.7) |
| AID INH/RIF | 46/59 (78.0%; 65.3–87.7) | 63/66** (95.5%; 87.3–99.1) | 46/49 (93.9%; 83.1–98.7) | 63/76 (82.9%; 72.5–90.6) | 109/125 (87.2%; 80.0–92.5) |
| AID FQ/EMB | 46/59 (78.0%; 65.3–87.7) | 67/69 (97.1%; 89.9–99.6) | 46/48 (95.8%; 85.7–99.5) | 67/80 (83.8%; 73.8–91.1) | 113/128 (88.3%; 81.4–93.3) |
| AID AG | 46/58 * (79.3%; 66.6–88.8) | 59/63 *** (93.7%; 84.5–98.2) | 46/50 (92.0%; 80.8–97.8) | 59/71 (83.1%; 72.3–91.0) | 105/121 (86.8%; 79.4–92.2) |
| | **Sensitivity: Microscopy-positives (43)** | | **Sensitivity: Microscopy-negatives (16)** | | |
| AID INH/RIF | 40/43 (93.0%; 80.9–98.5) | | 6/16 (37.5%; 15.2–64.6) | | |
| AID FQ/EMB | 40/43 (93.0%; 80.9–98.5) | | 6/16 (37.5%; 15.2–64.6) | | |
| AID AG | 39/42 * (92.9%; 80.5–98.5) | | 7/16 (43.8%; 19.8–70.1) | | |

The diagnostic accuracy for *M. tuberculosis* detection compared to culture methods is depicted for each of the AID-modules in participants with culture-confirmed TB (59) and with No TB (69).

*One participant with microbiologically confirmed TB had an invalid M.tb result for the AG module.

**Three participants with no TB had an invalid *M. tuberculosis* result for the INH/RIF module.

***Six participants with no TB had an invalid M.tb result for the AG module.

****Five participants had a false positive M.tb result for any of the three AID modules: one participant was also Xpert MTB/RIF-positive, three participants reported to have been exposed to TB previously, three participants were treated for TB in the past, one participant was diagnosed with pulmonary TB Sequelae as cause of current symptomatic episode, two participants were diagnosed with pneumonia (one interstitial, one lobar) as cause of current symptomatic episode.

The diagnostic sensitivity for *M. tuberculosis* detection compared to culture methods is depicted for each of the AID-modules in participants with culture-confirmed TB, stratified by smear microscopy status of the TB patients. *One participant with microbiologically confirmed TB had an invalid *M. tuberculosis* result for the AG module.

**Table 3. Resistance detection by AID-modules versus DST in culture as reference standard.**

| AID drug/locus | No of subjects included* | Sensitivity | Specificity | PPV | NPV | Agreement/AUC |
|---|---|---|---|---|---|---|
| AID RIF | 32 | 3/3, 100% (29.2–100.0) | 29/29, 100% (88.1–100.0) | 3/3, 100% (29.2–100.0) | 29/29, 100% (88.1–100.0) | 32/32, 100% (89.1–100.0) |
| AID INH | 32 | 2/2, 100% (15.8–100.0) | 29/30, 96.7% (82.8–99.9) | 2/3, 66.7% (9.4–99.2) | 29/29, 100% (88.1–100.0) | 31/32, 96.9% (83.8–99.9) |
| AID FQ | 10 | 1/1, 100% (2.5–100.0) | 9/9, 100% (66.4–100.0) | 1/1, 100% (2.5–100.0) | 9/9, 100% (66.4–100.0) | 10/10, 100% (69.2–100.0) |
| AID EMB | 17 | - | 17/17, 100% (80.5–100.0) | - | 17/17, 100% (80.5–100.0) | 17/17, 100% (80.5–100.0) |
| AID STR | 37 | 2/3, 66.7% (9.4–99.2) | 34/34, 100% (89.7–100.0) | 2/2, 100% (15.8–100.0) | 34/35, 97.1% (85.1–99.9) | 36/37, 97.3% (85.8–99.9) |
| AID KAN/AMK | 41 | 0/1, 0.0% (0.0–97.5) | 39/40, 97.5% (86.8–99.9) | 0/1, 0.0% (0.0–97.5) | 39/40, 97.5% (86.8–99.9) | 39/41, 95.1% (83.5–99.4) |
| AID KAN/AMK/CAP | 43 | - | 43/43, 100% (91.8–100.0) | - | 43/43, 100% (91.8–100.0) | 43/43, 100% (91.8–100.0) |

*Only participants with available results for the respective AID module and culture DST were included. DST culture results were missing for three M.tb-positive participants. No AID resistance-information available, due to false *M. tuberculosis*-negative AID result, in 13 participants for the INH/RIF module (all INH/RIF sensitive by DST in culture), in 13 participants for the FQ/EMB module (13 FQ-sensitive and 12 EMB-sensitive by DST in culture, see Table 4, below) and in 12 participants for the AG-module (all AG-sensitive by DST in culture). No AID resistance-information available, due to an invalid AID resistance result, in 13 participants for INH and for RIF, in 28 participants for EMB and 36 participants for FQ, and nine, five and three participants for STR, KAN/AMK and KAN/AMK/CAP, respectively. For more details on invalid AID tests, refer to S1 Table in S3 File.

in DST and sequencing, but could not be assessed by the AID INH/RIF module due to an invalid test result.

There were four *M. tuberculosis* strains, which showed resistance to aminoglycosides in culture DST, while one strain resistant against streptomycin (2/3, 66.7%, 95%CI: 9.4–99.2) and another strain with resistance against kanamycin (0/1, 0.0%, 95%CI: 0.0–97.5) were not detected by the AID-LPA. Further, there was a strain from one participant, in whom a resistance to kanamycin and amikacin (rrs A1400G) was incorrectly detected by the AID LPA, resulting in a specificity of 97.5% (95%CI: 86.8–99.9). There was another participant infected with a fluoroquinolone-resistant (GyrA D94Y) *M. tuberculosis* strain, who was flagged resistant by all three testing methods, (**Tables 3 and 4**).

In summary, there were three patients with MDR-TB included in this study, all of them were correctly identified as rifampicin-resistant by the AID-LPA, while one out of three had an invalid isoniazid-result. Among the high proportion of patients, who were not included in this evaluation due to a false negative *M. tuberculosis* results or an invalid result by the AID-LPA modules, no relevant mutations were detected by culture DST and sequencing.

## Invalid AID-tests

In our hands, the AID-tests produced a high number of invalid results. In **S1 Table in S3 File**, the distribution of invalid AID-LPA results is depicted for each module and in several subgroups. In this study, invalid results for *M. tuberculosis* detection were more likely in participants classified as "No TB" than in participants with culture-confirmed TB. Also, the number of invalid results from participants with detectable acid-fast bacilli on sputum smear microscopy was greater compared to smear microscopy-negative participants. In general, the majority of invalid tests was obtained for resistance testing and not *M. tuberculosis* detection, whereby the FQ/EMB- module was most affected (78.26% for FQ/EMB versus 32.61% for INH/RIF and 19.57% for AG of invalid resistance testing results).

**Table 4. Pattern of resistance results by all three tests: sequencing, AID modules and DST in culture.**

| Drug | Gene locus | SEQ** | AID | DST | Subject ID | Interpretation |
|---|---|---|---|---|---|---|
| Rifampicin | RIF rpoB S450L | R | R | R | 055-E | MDR-TB |
| | RIF rpoB S450L | R | R | R | 060-N | MDR-TB |
| | RIF rpoB S450L | no data | R | R | 118-B | MDR-TB |
| Isoniazid | INH katG S315T | R | R | R | 055-E | MDR-TB |
| | INH katG S315T | R | **invalid** | R | 060-N | MDR-TB |
| | INH katG S315T | R | R | S | 103-R | Isolated INH-resistance |
| | INH katG S315T, inhA -16, -15, -8 | no data | R | R | 118-B | MDR-TB |
| | INH inhA L217V* | nonWT | **invalid** | S | 128-M | No INH-resistance |
| Gyrase-inhibitors | FQ GyrB T373T* | nonWT | **invalid** | S | 063-U | No FQ- resistance |
| | FQ GyrA D94Y | R | R | R | 111-R | Isolated FQ-resistance |
| Ethambutol | EMB embB T1027T* | nonWT | **invalid** | S | 001-B | No EMB-resistance |
| | EMB embC S1022S* | nonWT | **invalid** | S | 020-P | No EMB-resistance |
| | N/A | gWT | **M.tb neg** | R | 030-U | No EMB-resistance |
| | EMB embA S1022S* | nonWT | **invalid** | S | 084-R | No EMB-resistance |
| | EMB embC S1022S* | nonWT | **invalid** | S | 114-C | No EMB-resistance |
| Aminoglycosides | STR rpsL K43R | R | R | R | 055-E | AG-resistance |
| | STR rpsL K43R | R | **S** | R | 060-N | AG-resistance |
| | STR rpsL A88G | no data | R | R | 118-B | AG-resistance |
| | KAN/AMK rrs A1400G | gWT | **R** | S | 004-U | No AG-resistance |
| | Kanamycin | no data | S | R | 118-B | AG-resistance |

Resistance data for all three tests are shown for participants with any drug resistance (R) by culture DST, AID assay or sequencing <u>and</u> for patients with an invalid AID-test result plus nonWT mutation for the respective drug by sequencing. Those with a not-evaluable AID result but WT und S results in DST were not included in this table; *these regions are not covered by AID-test; invalid = absence of wt and mutation bands for respective genome region, SEQ = sequencing, AID = AID line probe assay, DST = Drug Susceptibility Testing in culture, R = resistant, S = sensitive, gWT = genomic Wild Type, nonWT = non Wild Type.

## Discussion

Rapid detection of active TB and *M. tuberculosis* drug resistance is a prerequisite for immediate and correct TB treatment initiation in order to reduce TB morbidity and improve TB control. In our study we prospectively evaluated the diagnostic accuracy of three modules of the new AID-LPA, which included the diagnosis of *M. tuberculosis* and the detection of first and second line drug resistance.

As the study was performed on sputum samples of clinically symptomatic patients with presumed TB, the capacity of all three test modules to detect *M. tuberculosis* was analyzed in a first step. While the diagnostic capacity in participants with detectable acid-fast bacilli on sputum smear microscopy was good with about 93%, it was poor with about 40% in patients with paucibacillary disease. A difference—although not of this size—in the diagnostic performance depending on the mycobacterial load was expected and is already accepted knowledge for other molecular *M. tuberculosis* tests such as Hain assays [15,16] and even assays using the GeneXpert platform [17]. Further, systematic reviews and meta-analyses showed a wide range in sensitivity ranging from 40% to over 90% for molecular line probe assays, also depending on the study population, the underlying prevalence of drug resistance, the study laboratory and the study design [5,15,16]. Unfortunately, the reduced capacity of the AID-LPA assay to detect *M. tuberculosis* as a prerequisite for downstream resistance evaluation prevented more than 20% of the study participants with a false negative *M. tuberculosis* result from the inclusion in the drug resistance analysis.

Apart from false-negative *M. tuberculosis* results, also the high number of non-evaluable test results as well as the very low prevalence of drug resistance in our study population limited the evaluation of resistance testing of all AID modules. Therefore, our results need to be interpreted with caution and are not generalizable. In general, if a valid result could be obtained, the performance of the RIF/INH and FQ/EMB modules was very good. There were three MDR-TB patients in our study, all of them were correctly identified as rifampicin resistant, while one had a non-evaluable result for isoniazid. Interestingly, one participant, who was found isoniazid-sensitive by culture DST, was correctly identified (confirmed by sequencing) as an isoniazid-resistant TB case by the AID test. Opposed to that, the performance of the aminoglycoside resistance-module was only moderate; two resistant cases (one streptomycin and one kanamycin) were not detected by the AID test, while one kanamycin-resistance was incorrectly diagnosed in another participant, which could neither be confirmed by DST in culture nor by sequencing. Thus, based on our limited data, it could be hypothesized that the AID-LPA could be useful to identify MDR-TB cases early before treatment start or phenotypic resistance results are available. General conclusions regarding second line resistance testing in identified rifampicin/isoniazid-resistant cases cannot be drawn based on limited sample size and as both, false negative and false positive second line resistance results were observed in our study. In the two previous evaluation studies either no [6] or only a small amount [7] of strains resistant against second line drugs were included. Contrary to our findings, in the study by Molina-Moya et al. [7] the AG-module showed a very good and the FQ/EMB module only a poor to moderate diagnostic performance. In any case, the available data from all evaluation studies are not sufficient to propose a specific second line resistance testing strategy including any of the AID-LPA modules. Therefore, any AID-LPA results warrant confirmation by phenotypic resistance testing.

The high proportion of non-evaluable test results for all AID modules was striking and reasons for that remain unclear. Although a second test run was performed if a strip showed any non-evaluable result, the number of valid tests could not be substantially increased. Also, the implementation of modified testing protocols, which not only included amendments on the amplification protocol but also new lot numbers for reagents such as buffers and test kits, did not result in a reduction in the proportion of invalid test results. As the study was performed in an accredited TB laboratory, which was also experienced in conducting LPAs, it rather seemed that the AID testing modules were either not working robustly in the routine testing scenario of our study or are not suitable for sputum-based testing. Hain assays, which were performed in all samples in parallel to AID testing did not yield a high number of non-evaluable results (in total 8 invalid tests). Further, we assessed whether invalid results were associated with other mutations in the specific coding regions than those detected by the probes included in the AID modules. There were six strips with a non-evaluable result (one for INH/RIF- module, five for the FQ/EMB-module), where the corresponding sequencing data showed a non-wild type sequence. In all cases, however, the respective mutation was outside the target region covered by the AID testing probes and, thus, cannot explain the invalid AID result.

In conclusion, the current version of all AID modules have a moderate sensitivity and good specificity for the detection of *M. tuberculosis* in sputum samples from patients with presumed TB in Romania. However, the high proportion of invalid tests, especially with regards to resistance testing, is problematic and hampered the evaluation in this study.

## Supporting information

**S1 File.**
(TXT)

**S2 File.**
(DOCX)

**S3 File.**
(DOCX)

## Acknowledgments

We are grateful to all study participants for their participation and wish to acknowledge the study teams at MNI in Bucharest, at the Division for Infectious Diseases and Tropical Medicine at LMU and the Research Center Borstel. We thank Dr. Rosemarie Preyer and Dr. Volker Haid for technical support.

## Author Contributions

**Conceptualization:** Andrea Rachow, Stefan Niemann, Ioana D. Olaru, Michael Hoelscher, Christoph Lange, Elmira Ibraim.

**Formal analysis:** Andrea Rachow, Elmar Saathoff, Matthias Merker, Sabine Kastner.

**Funding acquisition:** Andrea Rachow, Stefan Niemann, Ioana D. Olaru, Michael Hoelscher, Christoph Lange, Elmira Ibraim.

**Investigation:** Roxana Mindru, Oana Popescu, Doinita Lugoji, Matthias Merker, Stefan Niemann, Sabine Kastner, Elmira Ibraim.

**Methodology:** Andrea Rachow, Elmar Saathoff, Michael Hoelscher, Elmira Ibraim.

**Project administration:** Andrea Rachow, Beatrice Mahler, Elmira Ibraim.

**Supervision:** Andrea Rachow, Roxana Mindru, Sabine Kastner.

**Writing – original draft:** Andrea Rachow.

**Writing – review & editing:** Elmar Saathoff, Roxana Mindru, Oana Popescu, Doinita Lugoji, Beatrice Mahler, Matthias Merker, Stefan Niemann, Ioana D. Olaru, Sabine Kastner, Michael Hoelscher, Christoph Lange, Elmira Ibraim.

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
