## [Decision Letter · Decision Letter 0]

9 Mar 2022

PONE-D-21-35811Diagnostic performance of the AID Line Probe Assay in the detection of Mycobacterium tuberculosis and drug resistance in Romanian patients with presumed TBPLOS ONE

Dear Dr. Ibraim

Thank you for submitting your manuscript to PLOS ONE. After careful consideration, we feel that it has merit but does not fully meet PLOS ONE’s publication criteria as it currently stands. Therefore, we invite you to submit a revised version of the manuscript that addresses the points raised during the review process.

We look forward to receiving your revised manuscript.

Kind regards,

Pradeep Kumar, Ph.D.

Academic Editor

PLOS ONE

Journal Requirements:

2. Please include a complete copy of PLOS’ questionnaire on inclusivity in global research in your revised manuscript. Our policy for research in this area aims to improve transparency in the reporting of research performed outside of researchers’ own country or community. The policy applies to researchers who have travelled to a different country to conduct research, research with Indigenous populations or their lands, and research on cultural artefacts. The questionnaire can also be requested at the journal’s discretion for any other submissions, even if these conditions are not met.  Please find more information on the policy and a link to download a blank copy of the questionnaire here: https://journals.plos.org/plosone/s/best-practices-in-research-reporting. Please upload a completed version of your questionnaire as Supporting Information when you resubmit your manuscript

Reviewers' comments:

Reviewer's Responses to Questions

**Comments to the Author**

1. Is the manuscript technically sound, and do the data support the conclusions?

Reviewer #1: Partly

2. Has the statistical analysis been performed appropriately and rigorously? 

Reviewer #1: Yes

3. Have the authors made all data underlying the findings in their manuscript fully available?

Reviewer #1: Yes

4. Is the manuscript presented in an intelligible fashion and written in standard English?

Reviewer #1: No

5. Review Comments to the Author

Reviewer #1: Development of rapid drug resistance for Tb is an urgent need in order to control Tb transmission globally. The authors have sought to evaluate the performance on a new LPA, using p-DST and sequencing as reference standard. The reviewer agrees with conclusions of the authors AID test has moderate sensitivity and high specificity for Tb diagnostic, but due to high proportion of non-evaluable tests would not be able to conclude the assay has high drug resistance sensitivity (the confidence intervals are extremely wide).

Line 134: reference missing

Table 2 requires revision. It is not very intuitive. It is not clear what is the comparative assay for “All participants” section. I wonder if it is better to separate the table into table 2A where the comparative assay is phenotypic drug susceptibility assay and then table 2B where the comparative assay will be smear microscopy.

Line 231: is non-evaluable term coined by the manufacturer and commonly used within diagnostic practices? Would “indeterminate” be a better alternative? It would certainly help the reader if non-evaluable is defined in the methods section. Moving the AID assay reader rules from supplementary section to main portion of the methods section is needed to understand the interpretation of the assay output. In the methods section the authors describe “invalid” results. Is it same as non-evaluable?

Line 237: is it table 4?

Line 241: inhA -16?

Table 5: is not very intuitive as well. It seems like the authors are trying to determine whether the non-evaluable results is associated with smear status. It would help to discuss these numbers of non-evaluable results in the results section and move the table to the supplementary section. Additionally, if the authors could comment on whether they investigated for an association with a lot number of the kits, instruments, or buffer.

6. PLOS authors have the option to publish the peer review history of their article (what does this mean?). If published, this will include your full peer review and any attached files.

Reviewer #1: No

---

## [Author Response · Author response to Decision Letter 0]

20 Jun 2022

Dear Editor and Reviewers!

We are very grateful for the thorough and thoughtful way in which this manuscript has been reviewed. We are pleased that we were given the chance to submit a revised version of our manuscript. We hope that we have correctly addressed all comments and that the changes have improved the manuscript significantly.

Please find a point-by-point response to the helpful comments below:

Comment #1 of Reviewer #1: Development of rapid drug resistance for Tb is an urgent need in order to control Tb transmission globally. The authors have sought to evaluate the performance on a new LPA, using p-DST and sequencing as reference standard. The reviewer agrees with conclusions of the authors AID test has moderate sensitivity and high specificity for Tb diagnostic, but due to high proportion of non-evaluable tests would not be able to conclude the assay has high drug resistance sensitivity (the confidence intervals are extremely wide).

Authors` reply to Comment #1 of Reviewer #1: Thank you for the thorough evaluation of our manuscript and your guidance. We are pleased that the reviewer agrees with our main conclusions regarding the diagnostic accuracy of the AID test with respect to the detection of M. tuberculosis and the prediction of anti-tuberculosis drug-resistance.

Comment #2 of Reviewer #1: Line 134: missing reference

Authors` reply to Comment #2 of Reviewer #1: Thank you very much for noticing this mistake. The WHO technical guidelines to which we are referring to is reference number “15” as it was already indicated at the end of the sentence (line 136, in manuscript with track-changes). Thus, we have removed “(REF)” from the manuscript (line 136, in manuscript with track-changes). 

Comment #3 of reviewer #1: Table 2 requires revision. It is not very intuitive. It is not clear what is the comparative assay for “All participants” section. I wonder if it is better to separate the table into table 2A where the comparative assay is phenotypic drug susceptibility assay and then table 2B where the comparative assay will be smear microscopy.

Authors` reply to Comment #3 of Reviewer #1: We agree with this comment and revised table 2 (incl. titles and legends) accordingly and separated it as suggested in 2A and 2B. As tables are different content-wise and from layout perspective, we would offer to change the numbering to table 2 and table 3, accordingly, and also that from all downstream tables. In table 2B we now calculate the sensitivity of AID-modules stratified by smear-microscopy status, in order to convey that sensitivity of the AID assay is greatly varying according to positive versus negative smear-microscopy status of the respective participant. We think that it is not appropriate in our study to use smear-microscopy as comparative assay, as smear microscopy is an insufficient reference standard with much lower sensitivity compared to culture methods, which would result in false positive AID-test results (i.e. in the 16 smear-negative but culture positive culture-confirmed TB patients) and may confuse some readers of the manuscript.

Comment #4 of reviewer #1: Line 231: is non-evaluable term coined by the manufacturer and commonly used within diagnostic practices? Would “indeterminate” be a better alternative? It would certainly help the reader if non-evaluable is defined in the methods section. Moving the AID assay reader rules from supplementary section to main portion of the methods section is needed to understand the interpretation of the assay output. In the methods section the authors describe “invalid” results. Is it same as non-evaluable?

Authors` reply to Comment #4 of Reviewer #1: We are grateful for this comment as it reflects previous internal discussion we had in our team. As the manufacturer does not account for missing or incomplete reaction zones in their manual, we searched the literature and found that for the analysis and reporting of other line probe test results (mainly Hain assays) the terms valid and invalid were majorly used (sometimes also the term indeterminate). Therefore we replaced the term non-evaluable by the term invalid. In addition, we provide a description of the term “invalid” in the methods section, lines 117-119, in manuscript version with trach-changes included.

Comment #5 of reviewer #1: Line 237: is it table 4?

Authors` reply to Comment #5 of Reviewer #1: Thanks a lot for spotting this mistake, we changed it to table 4 (line 253)

Comment #6 of reviewer #1: Line 241: inhA-16?

Authors` reply to Comment #6 of Reviewer #1: Thank you very much, your comment is correct. We added the dash accordingly (line 257).

Comment #7 of reviewer #1: Line Table 5: is not very intuitive as well. It seems like the authors are trying to determine whether the non-evaluable results is associated with smear status. It would help to discuss these numbers of non-evaluable results in the results section and move the table to the supplementary section. Additionally, if the authors could comment on whether they investigated for an association with a lot number of the kits, instruments, or buffer.

Authors` reply to Comment #7 of Reviewer #1: We agree with moving the table to the supplement. Further, we described the patterns of invalid results in different sub-groups in the result section, including modifications of the text in order to ensure that the message is understandable and clear, even without simultaneous viewing of the table. Additionally, we amended the discussion on invalid test results, including a sentence on the association with kit numbers, buffers or PCR-protocol (lines 367-370).

---

## [Editor Report · Decision Letter 1]

28 Jun 2022

Diagnostic performance of the AID Line Probe Assay in the detection of Mycobacterium tuberculosis and drug resistance in Romanian patients with presumed TB

PONE-D-21-35811R1

Dear Dr. Rachow,

We’re pleased to inform you that your manuscript has been judged scientifically suitable for publication and will be formally accepted for publication once it meets all outstanding technical requirements.

Kind regards,

Pradeep Kumar, Ph.D.

Academic Editor

PLOS ONE
---

## [Editor Report · Acceptance letter]

25 Jul 2022

PONE-D-21-35811R1 

Diagnostic performance of the AID Line Probe Assay in the detection of *Mycobacterium tuberculosis* and drug resistance in Romanian patients with presumed TB 

Dear Dr. Rachow:

I'm pleased to inform you that your manuscript has been deemed suitable for publication in PLOS ONE. Congratulations! Your manuscript is now with our production department. 

Kind regards, 

on behalf of

Dr. Pradeep Kumar 

Academic Editor

PLOS ONE